# The Role of Lung Ultrasound in SARS-CoV-19 Pneumonia Management

**DOI:** 10.3390/diagnostics12081856

**Published:** 2022-07-31

**Authors:** Marina Lugarà, Stefania Tamburrini, Maria Gabriella Coppola, Gabriella Oliva, Valeria Fiorini, Marco Catalano, Roberto Carbone, Pietro Paolo Saturnino, Nicola Rosano, Antonella Pesce, Raffaele Galiero, Roberta Ferrara, Michele Iannuzzi, D’Agostino Vincenzo, Alberto Negro, Francesco Somma, Fabrizio Fasano, Alessandro Perrella, Giuseppe Vitiello, Ferdinando Carlo Sasso, Gino Soldati, Luca Rinaldi

**Affiliations:** 1U.O.C. Internal Medicine, ASL Center Naples 1, P.O. Ospedale del Mare, 80147 Naples, Italy; gabry.cop@libero.it (M.G.C.); gably@libero.it (G.O.); 2U.O.C. Radiology, ASL Center Naples 1, P.O. Ospedale del Mare, 80147 Naples, Italy; tamburrinistefania@gmail.com (S.T.); valeria.fiorini@libero.it (V.F.); marco26catalano@yahoo.it (M.C.); robcarbone@alice.it (R.C.); pietropsaturnino@libero.it (P.P.S.); nicola.rosano89@gmail.com (N.R.); antonellapesce1986@libero.it (A.P.); 3Department of Advanced Medical and Surgical Sciences, University of Campania Luigi Vanvitelli, 80121 Naples, Italy; raffaele.galiero@gmail.com (R.G.); roberta.ferrara@libero.it (R.F.); ferdinandocarlo.sasso@unicampania.it (F.C.S.); lucarinaldi@hotmail.it (L.R.); 4Department of Anesthesia and Intensive care Medicine, ASL Center Naples 1, P.O. Ospedale del Mare, 80147 Naples, Italy; michele.iannuzzi@aslnapoli1centro.it; 5U.O.C. Neurodiology, ASL Center Naples 1, P.O. Ospedale del Mare, 80147 Naples, Italy; vincenzo-dagostino@libero.it (D.V.); alberto.negro@hotmail.it (A.N.); fra1585@hotmail.it (F.S.); fabriziodoc@gmail.com (F.F.); 6Infectious Diseases at Health Direction, AORN A. Cardarelli, 80131 Naples, Italy; alessandro.perrella@aocardarelli.it; 7Healt Direction, ASL Center Naples 1, P.O. Ospedale del Mare, 80147 Naples, Italy; giuseppe.vitiello@aslnapoli1centro.it; 8Diagnostic and Interventional Ultrasound Unit, Valle del Serchio General Hospital, Castelnuovo Garfagnana, 55032 Lucca, Italy; soldatigino@yahoo.it

**Keywords:** lung ultrasound, high resolution computed tomography, SARS-CoV-19, interstitial pneumonia, ARDS

## Abstract

Purpose: We aimed to assess the role of lung ultrasound (LUS) in the diagnosis and prognosis of SARS-CoV-2 pneumonia, by comparing it with High Resolution Computed Tomography (HRCT). Patients and methods: All consecutive patients with laboratory-confirmed SARS-CoV-2 infection and hospitalized in COVID Centers were enrolled. LUS and HRCT were carried out on all patients by expert operators within 48–72 h of admission. A four-level scoring system computed in 12 regions of the chest was used to categorize the ultrasound imaging, from 0 (absence of visible alterations with ultrasound) to 3 (large consolidation and cobbled pleural line). Likewise, a semi-quantitative scoring system was used for HRCT to estimate pulmonary involvement, from 0 (no involvement) to 5 (>75% involvement for each lobe). The total CT score was the sum of the individual lobar scores and ranged from 0 to 25. LUS scans were evaluated according to a dedicated scoring system. CT scans were assessed for typical findings of COVID-19 pneumonia (bilateral, multi-lobar lung infiltration, posterior peripheral ground glass opacities). Oxygen requirement and mortality were also recorded. Results: Ninety-nine patients were included in the study (male 68.7%, median age 71). 40.4% of patients required a Venturi mask and 25.3% required non-invasive ventilation (C-PAP/Bi-level). The overall mortality rate was 21.2% (median hospitalization 30 days). The median ultrasound thoracic score was 28 (IQR 20–36). For the CT evaluation, the mean score was 12.63 (SD 5.72), with most of the patients having LUS scores of 2 (59.6%). The bivariate correlation analysis displayed statistically significant and high positive correlations between both the CT and composite LUS scores and ventilation, lactates, COVID-19 phenotype, tachycardia, dyspnea, and mortality. Moreover, the most relevant and clinically important inverse proportionality in terms of P/F, i.e., a decrease in P/F levels, was indicative of higher LUS/CT scores. Inverse proportionality P/F levels and LUS and TC scores were evaluated by univariate analysis, with a P/F–TC score correlation coefficient of −0.762, *p* < 0.001, and a P/F–LUS score correlation coefficient of −0.689, *p* < 0.001. Conclusions: LUS and HRCT show a synergistic role in the diagnosis and disease severity evaluation of COVID-19.

## 1. Introduction

COVID-19 is an infectious disease with a wide range of clinical symptoms, ranging from asymptomatic to mildly symptomatic and severe forms, pointing to a major role of the host response to SARS-CoV-2 (severe acute respiratory syndrome coronavirus 2) [1]. The clinical spectrum of SARS-CoV-2 infection is broad, ranging from asymptomatic infection to flu-like illness, to severe and diffuse viral pneumonia with a life-threating course, related to cytopathic and immune-mediated injury in the pulmonary parenchymal. Patients may show symptoms that include fever, high temperature, cough, myalgia, sputum production, headache, hemoptysis, diarrhea, dyspnea, and, in some cases, acute respiratory distress syndrome (ARDS), acute cardiac injury, or secondary infection. Most of infections are not severe, 81% are mild, 14% of the cases are severe (with dyspnea, hypoxia, or >50% lung involvement on diagnostic imaging), and 5% develop a critical disease with respiratory failure, shock, or multiorgan dysfunction [2]. The risk of death from COVID-19 strongly depends on the patient’s age and previous health status. Older patients are much more prone to critical and fatal disease outcomes, especially with comorbidities, including cardiovascular diseases, hypertension, chronic kidney disease, diabetes, and pulmonary disease [3]. Thrombotic microangiopathy and complement activation, pulmonary embolism, and elevated D-dimer levels have also been reported with high frequency in patients with COVID-19 [4,5,6,7]. Numerous previous studies, including the paper by Giannini and al. [8,9], have already discussed the significance of the D-dimer level as an independent predictor of mortality in severe cases of ARDS during SARS-CoV-2. COVID-19 laboratory diagnosis is based on real-time polymerase chain reaction (RT-PCR) assay obtained by oro-nasopharyngeal swab sample, bronchoalveolar lavage, or tracheal aspirate, while imaging plays a major role in the early diagnosis of the pleuropulmonary complications [8,9,10]. The pathophysiology of severe COVID-19 infection is marked by elevated numbers of neutrophils in the nasopharyngeal epithelium, in the distal parts of the lungs, and in blood. The experience gained during the Italian epidemic pointed to patients’ age as one of the most important risk factors for COVID-19 mortality [11]. However, a recent study demonstrated that patients who died of COVID-19 appear to have lost considerable lifetime, independent of their age. Imaging findings significantly support clinical judgement to ensure effective and timely management and prognosis; indeed, the identification of disease severity allows appropriate selection for early involvement of intensive care [10,11,12]. Contrary to X-ray, chest computed tomography (CT) plays a pivotal role in the diagnosis and monitoring of interstitial pneumonia [13,14]. Typical CT patterns of COVID-related pneumonia include multifocal bilateral peripheral ground glass opacities associated with subsegmental patchy consolidations, commonly subpleural and predominantly involving lower lung lobes and posterior segments [15,16]. Likewise, lung ultrasound (LUS) shows certain advantages for detecting and monitoring COVID-19 “pneumonia” [17]. This diagnostic technique is safe, repeatable, and can be used with low cost at the bedside in absence of radiation exposure [16,17,18]. Moreover, it is useful for rapid assessment of the severity of SARS-CoV-2 pneumonia/ARDS (acute respiratory distress syndrome) in diagnosis and follow-up settings, and for monitoring lungs during recruitment maneuvers and in prone positions [19,20]. The use of LUS for patients with suspected COVID-19 may reduce the risk associated with transporting unstable patients to CT rooms, which is especially important for preventing nosocomial outbreaks due to high contagiousness of virus [21,22].

Thoracic ultrasound has been employed for the diagnosis of many thoracic diseases and is an accepted detection tool for pleural effusions, atelectasis, pneumothorax, and pneumonia. However, the use of ultrasound for the evaluation of parenchymal lung disease, when the organ is still aerated, is a relatively new application. The diagnosis of a normal lung and the differentiation between a normally aerated lung and a lung with interstitial pathology are based on the interpretation of ultrasound artifacts universally known as A- and B-Lines. Even though the practical role of lung ultrasound artifacts is accepted by many clinicians, their physical basis and the correlations between these signs and the causal pathology is not understood in detail [23]. The utility of a lung ultrasound (LUS) in the diagnosis of interstitial lung disease (ILD in very early SSc has also been described including, more recently, its potential for the detection of SSc-ILD in asymptomatic preclinical stages. Recent research has focused on the predictive value of LUS [24,25], which is promising for the application of LUS as a screening method for SSc-ILD in clinical practice. Although these are strong arguments in favor of the application of LUS in SSc, to date there is no unanimous consensus on the role LUS plays in the diagnosis and/or prognosis of SSc-ILD [26].

The use of LUS for patients with suspected COVID-19 may reduce the risk associated with transporting unstable patients to CT rooms, which is especially important for preventing nosocomial outbreaks, due to the high contagiousness of the virus. The purpose of our study was to determine the role of LUS in the diagnosis and prognosis of SARS-CoV-2 pneumonia, considering high-resolution computed tomography (HRTC) as the gold standard.

## 2. Materials and Method

### 2.1. Patients

From March 2020 to October 2020, all patients who had been diagnosed with COVID-19 infection by RT-PCR on nasopharyngeal swab samples and throat swabs, and subsequently hospitalized at our COVID Centre, were enrolled in the study. Patients underwent LUS and CT within 48–72 h of admission to our emergency department. Anamnestic, epidemiological, and demographic data were collected either from the patients themselves or from their families, and were recorded. All the results, clinical and laboratory data, and pulmonary CT and LUS were analyzed retrospectively and aggregated anonymously. Comorbidities and related therapies, including obesity, chronic kidney disease, hypertension, type 2 diabetes mellitus, atrial fibrillation, coronary artery disease, dementia, chronic obstructive pulmonary disease (COPD), chronic hepatitis, history of cancer in the last 5 years, and smoking were also recorded.

Clinical phenotypes were classified into four groups as follows:

1. pauci-symptomatic subjects (fever, no hypoxemia); 

2. mildly symptomatic patients (fever, mild hypoxemia with pO_2_ 40–60 mmhg, need of oxygen therapy with nasal cannula and vent-mask); 

3. moderately symptomatic patients (fever, moderate to severe respiratory failure with pO_2_ < 40 mmhg, need of CPAP/NIV);

4. patients with severe disease (severe respiratory failure with pO_2_ < 40 mmhg, ARDS, with or without invasive ventilation).

### 2.2. Statement of Ethics

The study protocol was approved by the Institutional Ethics Committee of Ancona (ID 0179104/I, 28 July 2021). The study was conducted in accordance with the principles of the Declaration of Helsinki. Due to the retrospective design of the study and anonymous collection of data, informed consent signature was waived, in compliance with Italian law.

### 2.3. Laboratory Data

Venous blood samples were collected from all patients to assess complete blood counts, with differentiation, fibrinogen, D-dimer, NT pro-bnp, troponin, and biochemistry tests (creatinine, LDH, C-reactive protein, ferritin, procalcitonin, glycemia, lymphocytes, PT, PTT, INR).

All patients underwent blood gas analysis on admission, by radial artery cannulation or puncture. The oxygenation status was assessed by O_2_ partial pressure value (pO_2_), CO_2 partial_ pressure value (pCO_2_) and hemoglobin oxygen saturation (SO_2_). We used the P/F ratio to compare different values of arterial pO_2_ in patients receiving different fractions of inspired oxygen (FiO_2_) by non-invasive ventilation. PaO_2_/FiO_2_ (P/F) ratio was used to classify the severity of ARDS, according to the Berlin definition, even though most of the evidence derived from intensive care settings.

According to the 2012 Berlin definition by the ARDS Definition Taskforce, a ratio of the partial pressure of arterial oxygen (PaO_2_) to the fraction of inspired oxygen (FiO_2_) (P/F ratio) of ≤100, 101–200 or 201–300 mmHg is deemed as severe, moderate, or mild, respectively.

Lactate levels were also recorded to evaluate any possible degree of tissue perfusion.

### 2.4. Oxygen Requirement

Ventilatory support was categorized into three groups: nasal cannula, Venturi mask and, non-invasive ventilation (CPAP and b-pap). Patients admitted to intensive care required tracheal intubation. 

### 2.5. LUS Protocol

Lung ultrasound examinations were performed at the bedside by trained sonographers (three internists and two radiologists) using portable ultrasound machines (Alpinion E cube i7 and Mindray DP10), equipped with a convex probe (3.5 MHz) and a linear probe (7.5–10 MHz). No harmonic filter was used. The linear and convex probes were each used in every patient. A reduced mechanical index was used as often as possible to obtain interpretable images. The depth was set to provide a clear view of the pleural line and 3–4 cm of the field below it. The gain was set in the intermediate position. The focus was placed at the level of the pleural line.

All the devices, the US scanner, probes, and cables, were wrapped in single-use plastic covers to reduce the risk of contamination and to facilitate the sterilization procedures.

#### Lung Ultrasound Scoring System 

The thorax was explored in twelve areas, six on each hemithorax, by intercostal scans (Figure 1). In critically ill patients who could not maintain the sitting position, paravertebral scans were acquired and moved as posteriorly as possible and towards the lowest and apical points, by placing the patient in an oblique position. Each area was assessed with the probe perpendicular to the chest wall, searching for the following signs: pleural line (a horizontal hyperechoic line between the ribs), A-lines (horizontal replica artifacts repeated at a constant distance equal to the distance between the pleural line and probe surface), vertical artifacts (vertical hyperechoic artefacts spreading from the pleural line towards the bottom of the screen), white lung (focal or multifocal artifacts characterized by an undifferentiated echogenic background, with the absence of A-lines and without evidence of vertical artifacts), and consolidation (presence of a tissue-like pattern) [23,24,25].

A standardized and COVID-19-dedicated four-level scoring system was used to categorize the ultrasound examination, according to published papers [26,27]: 

Score 0: The pleural line is regular. Horizontal reverberant artifacts (A-Lines) and mirror effects are present. Absence of visible alterations with ultrasound.

Score 1: The pleural line has slight alterations with sporadic vertical bright artifacts. These are often grouped and separated by the absence of visible alterations of the lung.

Score 2: The pleural line is broken at many points. Vertical artifacts are more numerous. Small subpleural consolidations may be present, often showing a cuneiform shape.

Score 3: The pleural line is irregular and cobbled. The subpleural lung is denser and more disordered. White lung with or without larger consolidations may be present. Small and large consolidations are evident in the scanned parts of the lung. 

For each patient, the total score was computed by adding the scores for each area explored. The total scores ranged from 0 (best) to 36 (worst).

Severity of US pulmonary involvement was classified as mild (1–5), moderate (>5–15), or severe (>15). 

The presence of pleural effusion and lung sliding was recorded. Ultrasound images and videos were stored and numbered from the right posterior basal regions. 

### 2.6. High Resolution CT (HRCT) 

All patients had an initial chest HRCT scan within 48 h of hospitalization, using a multidetector 64-channel CT machine (Toshiba Aquilion PRIME). The detailed parameters for CT acquisition were as follows: tube voltage, 120 kVp; tube current, standard (reference mAs, 60–120); slice thickness, 1.0 mm; reconstructed interval, 1.0 mm. Patients were placed in a supine position. To minimize motion artifacts, patients were instructed on breath-holding and images were acquired during a single breath-hold. The scanning range was from the apex to the lung base. All the images were stored in a picture archive and communication system. Image analysis was performed using the institutional digital database system.

Two radiologists with more than 10 years’ experience evaluated the images in consensus, to determine the disease pattern, distribution, stage, and severity score for each patient. When discordant, the final decision was reached collegially. No negative control cases were examined.

The scans were first assessed, whether negative or positive, for typical findings of COVID-19 pneumonia (bilateral, multi-lobar lung infiltration, posterior peripheral ground glass opacities) as defined by the RSNA consensus statement and peer-reviewed literature on viral pneumonia. Recorded findings included ground glass opacity (GGO), crazy-paving pattern, and consolidation [16]. Atypical COVID-19 CT patterns were recorded.

CT findings were described as follows: (1) ground-glass opacities; (2) Consolidation; (3) ground-glass opacity with consolidation (assessing their respective predominance); (4) crazy-paving patterns; (5) no abnormalities. 

Ground-glass opacification was defined as hazy increased lung attenuation with preservation of bronchial and vascular margins, and consolidation was defined as opacification with obscuration of vessel margins and airway walls. Crazy paving represented GGO opacity with superimposed inter- and intralobular septal thickening [10,11]. Consolidation consisted of parenchyma deprived of air.

For each pulmonary lobe (five lobes), the volumetric parenchymal involvement was estimated with a score system as follows: 0 = no involvement; 1 = <5% involvement; 2 = 5–25% involvement; 3 = 26–49% involvement; 4 = 50–74% involvement; 5 = >75%.

The total CT score was the sum of the individual lobar scores, and ranged from 0 (no involvement) to 25 (maximum involvement) [11,15,16,17,18,19,20,21,22,23,24,25,26,27,28]. A correlation between the total LUS score and a Pan score of 0 to 24 was shown to be significant in a previous work [29,30,31].

In addition, CT scans were evaluated according to their distribution, side, and lobe involvement predominance. Finally, the presence of underlying non-related lung disease such as emphysema or fibrosis was recorded.

### 2.7. End Points of the Study

The primary endpoint was to compare the lung ultrasound severity score and chest CT severity in COVID-19 clinical management. Furthermore, we wanted to explore the correlation between the imaging severity score and COVID-19 phenotypes, clinical and laboratory parameters, and oxygen requirements, thus establishing a prognostic role for lung ultrasound in COVID-19 patients.

### 2.8. Statistical Analysis

Categorical data were expressed as numbers and percentages, and continuous variables either as mean and standard deviation (SD) or median and interquartile range (IQR), according to their distribution, and were tested by the Shapiro–Wilk test.

The sample size was 79 patients for a confidence level of 95%.

The ultrasound thoracic score and the CT score were tested for correlation by a linear regression analysis, and a dispersion graph was reported to describe the degree of correlation. The respective correlations of the CT and ultrasound thoracic (US) scores with several outcomes were assessed, as well as the respiratory and laboratory parameters, by calculating the Spearman correlation coefficient. The ROC curve analysis was used to evaluate mortality as a function of the LUS score, and to identify an optimal cut-off value. A *p*-value < 0.05 was considered as statistically significant. All the analyses were performed by STATA software (StataCorp. College Station, TX: StataCorp LLC), version 15.5.

## 3. Results

Ninety-nine patients were included in the study, with a median age of 71 years (IQR 58–78 years); within the study group the incidence of SARS-CoV-2 was higher in males 68.7% than in females 31.3% (Table 1). All patients underwent a CT scan and a US scan.

Table 1 synthetizes the anthropometric, demographic, clinical, and biochemical characteristics of the study cohort.

The main symptoms on admission were fever and cough, detected in 90.9% of patients, and dyspnea in 89.9%, whilst asthenia was present in 76.8% of patients and pharyngeal hyperemia in 63.6%.

Table 2 illustrates the pre-existing diseases of the patients and their therapies, while Table 3 shows the patients’ phenotypes and their clinical evolution during their hospital stay. Among pre-existing diseases, the most prevalent was hypertension (85.9%), while chronic obstructive pulmonary disease (COPD) was present in more than half of the patients (56.6%). Hyper-inflammation syndrome, due to the excessive production of proinflammatory cytokines and the dysfunction of the immune response, was found in 84.8% of patients.

The clinical phenotype 3 was the most prevalent, which was characterized by fever, and moderate to severe respiratory failure, with pO_2_ < 40 mmhg and need of Nnon -invasive ventilation (CPAP/NIV).

Oxygen support was needed for 66% (65%) of patients, specifically nasal cannulas (CN) 7%, venturi mask (VM) 24%, high flows (HFNC) 10%, and non-invasive ventilation—biPap 15% and CPAP 10%.

The LUS and CT scores were analyzed for correlation with the mortality outcomes, coagulation, and respiratory parameters using the Spearman correlation coefficient. As reported in Table 3 and Table 4, the results were for both scores similar. Table 3 shows negative correlation of P/F and LUS score (r = −0.689), and pO_2_–LUS score (r = −0.486), and positive correlation of ventilation–LUS score (r = 0.562), and lactates–LUS score (r = 0.479). Table 4 shows negative correlation of P/F with TC score (r = −0.689) and pO_2_ (r = −0.470) and positive correlation with type of ventilation (0.530).

The LUS signs of COVID-19 in every patient were B-lines. The frequencies of irregular or blurred pleural lines were 45% and 15%, respectively. Sub-centimetric lung consolidation was seen in 18 patients (18%). 

The median ultrasound thoracic score was 28 (IQR 20–36) and the most frequent scores per single scan were 2 and 3 (47.6% and 32.2%, respectively) (Table 1). 

The mean CT score was 12.63 (SD 5.72), with most of the patients showing a score 2 pattern (59.6%).

Patients with clinical phenotypes 3 or 4 (71% of the patients enrolled) presented higher rates of bilateral involvement, with more involved zones, B-lines, pleural line abnormalities, and consolidation. 

The relationship between the LUS scores and the CT scores was assessed by a linear regression analysis and dispersion graph (Figure 2), showing a positive linear relationship between the two evaluation scoring systems, which despite not being high (rho = 0.352), did however reach statistical significance (*p* = 0.008) (Figure 3).

Using the Spearman correlation coefficient, use LUS and CT scores were analyzed for correlation with the mortality outcome, coagulation, and respiratory parameters. As reported in Table 3 and Table 4, the results were almost similar for both scoring systems. Table 3 showed negative correlation for P/F and LUS score (r = −0.689), and pO_2_–LUS score (r = −0.486), positive correlation for ventilation–LUS Score (r = 0.562), and lactates–LUS Score (r = 0.479). Table 4 showed negative correlation for P/F and TC score (r = −0.689), and pO_2_ (r = −0.470), and positive correlation by type of ventilation (0.530).

The bivariate correlation analysis displayed statistically significant and high positive correlations for the CT scores as well as the LUS scores with the following parameters: ventilation, lactates, COVID-19 phenotype, tachycardia, dyspnea, and mortality. 

The LUS scores showed a significant association with in-hospital mortality (OR 0.7, 0.95% CI 0.59–0.82) Table 5; *p* < 0.001), with the risk of invasive respiratory support increasing with a greater LUS score measured on patient arrival.

These positive correlations demonstrate that a higher composite LUS or CT score corresponded to a higher mortality rate and a more severe COVID-19 phenotype. Furthermore, there was a direct proportionality between increased CT and LUS scores on one hand and elevated lactate levels and a need for ventilation on the other. Remarkably lower correlations, though still statistically significant, were found for age and d-dimer.

## 4. Discussion

Lung CT is currently the standard for comparison of other imaging methods for anatomical definition down to the level of the secondary lung lobule.

The most important radiology societies [11,12,13,14,15,16] recommend the use of CT in the presence of moderate and severe features of COVID-19 when RT-PCR results are negative or not available, when there is high pre-test probability, and in the management of patients with worsening or severe respiratory symptoms.

The possibility of using ultrasound to evaluate pulmonary changes in COVID-19 has been proposed since the beginning of the pandemic [26]. Two factors indicated this possibility; first, the prevailing sub-cortical distribution of the injuries, necessary for their US visibility, shown by the Computed Tomography (CT); and second, the histopathology of COVID-19 pulmonary involvement, which generates different degrees of physical density (de-aeration) within the lung, ranging from diffuse patchy alveolar damage to consolidations. Subpleural hyper densities are detectable by ultrasound [30,31], and COVID-19 histopathology (expressed by hyperdensities) represents a measure of the injury of the respiratory tissue which, in the most severe cases, can lead to respiratory failure.

The most frequently observed anatomical feature of COVID-19 is diffuse alveolar damage (DAD), characterized by high levels of proinflammatory cytokines. Later, a proliferative phase develops, involving fibroblasts, myofibroblast, lymphocytes, and extracellular matrix, with intra-alveolar fibrin accumulation. Finally, large vessel thrombosis and microthrombi containing fibrin and platelets may be detected in arteries smaller than 1 mm. In short, ultrasound consolidation corresponds to a lung region deprived of air, therefore without gas exchange. Alternatively, consolidations can represent ischemic regions. This overlap of viral, inflammatory, immune, and vascular events is clinically expressed by disease phenotypes with different prognostic weight [32].

This non-homogeneity was highlighted by Gattinoni et al. [33]. They noted that COVID-19 subjects with respiratory failure showed different clinical patterns of “pneumonia” with different pathophysiology. The transition between different pulmonary COVID-19 phenotypes probably depends on the interaction of many factors, also affecting the ventilation modes and contributing to respiratory injury.

If ultrasound allows bedside assessment of the subpleural lung in terms of density, i.e., absolute or relative reduction of the air spaces and interstitial ratio, or de-aeration, it is reasonable to expect correlations between ultrasound signs and the clinical picture in patients with COVID-19 lung involvement [34]. 

Although LUS is not currently included in the main international guidelines for COVID-19 patient management, some authors have proposed semi-quantitative LUS scores, which can be used to quantify lung aeration [35]. This approach has recently been used in patients with COVID-19 pneumonia. [36,37].

In the light of recent evidence regarding the genesis of pulmonary signs in ultrasound [35], and their possible relationship with the superficial histopathology of the lung in terms of density and de-structuring, it can be theorized that ultrasound can help define a stratification of tissue damage in COVID-19 patients.

The four-level ultrasound score used in this paper is based on the concept of disease severity, with consideration given to extension of findings on the lung and the nature of tissue densities [38].

Despite the profound difference between CT (tomographic) and ultrasound (exploring only superficial densities), LUS and CT scores showed a weak (but statistically significant) positive linear relationship. This agrees with other observations [39,40,41]. A plausible explanation can be found in the prevalent peripheral, subpleural expression of COVID-19 pulmonary injuries [11,15], which can minimize the differences between the two methods (axial and superficial estimates).

Results of this study and other observations agree with this hypothesis. In practical terms, LUS can be considered an equally accurate alternative to CT in cases of COVID-19, particularly in situations where CT is not easily accessible or when molecular tests are not available [42,43].

In our study, a significant positive correlation was demonstrated between LUS score and CT score, and certain strategic parameters (COVID-19 phenotype, need for non-invasive ventilation, hematic lactate level, and mortality) (Table 3 and Table 4). As expected, P/F, pH, and pO_2_, displayed significant negative correlations. However, the most relevant and clinically important inverse proportionality was found in relation to P/F, i.e., higher LUS/CT scores were indicative of a decrease in P/F ratio.

This study reinforces the results presented by Perrone et al [37], which demonstrated a significant correlation between an equivalent US scoring system (of 12 fields), and established end points of clinical worsening, including high-flow oxygen support, ICU admission, and death. This is important, because their methodology of ultrasound exploration of the chest and the score used in their study (specifically proposed for COVID-19 patients) were the same as those we used.

Many other studies showed that the extent of lung abnormalities evaluated by the LUS score is a predictor of a worse outcome, ETI, or death [44,45]. However, the scoring systems used, and, above all, the number of regions explored are often not comparable to each other. For example, it has been assumed that a very marked reduction in the number of regions explored may decrease the accuracy of LUS, as demonstrated by Falster et al. [46] using the Mongodi score with analysis of only eight thoracic areas. 

From a practical point of view, our results justify the attribution of an LUS score to every COVID-19 patient, from the early stages of management and during monitoring in the various settings. LUS may have a potential role in Emergency department for triaging symptomatic patients, managing ventilation, weaning ICU patients, and monitoring COVID-19 pneumonia and its evolution toward ARDS in critically ill patients. Moreover, it may be considered a first-line alternative to chest X-ray and CT scan in critically ill patients.

Even though asymptomatic carriers may comprise 17.9–33.3% of patients with COVID-19 [47,48], and can contribute to the spread of the infection, the role of screening these asymptomatic carriers is not known. Ultrasonography could help identify infected people with signs and symptoms at the onset [49]. Compared to its sensitivity, the specificity of US to pathological artifacts of the lung in COVID-19 and other diseases is generally considered low.

US signs of COVID-19 are present in various degrees in other pathologies (diffuse pneumonia, pulmonary edema, interstitial lung diseases). The diagnostic accuracy of pulmonary ultrasound in COVID-19 is based on the diffuse and bilateral aspect of lung involvement, on the prevalence of artefactual components, and on a typically patchy distribution. Of course, the pretest probability of having contracted the infection assumes significance [34]. At present, to our knowledge, very few lung diseases with a large epidemic diffusion show these aspects. 

In this study, we found that the most common CT findings were GGO, consolidation, and crazy-paving patterns, including “spider web sign” (defined as a triangular or angular GGO in the subpleural lung with the internal interlobular septa thickened like a net). A differential diagnosis between COVID 19 and systemic sclerosis (SSc) with interstitial lung is made possible by employing CT images; the presence of consolidations and fibrosis inside GGO in the lower lobes are independent CT diagnostic features for COVID-19 [50]. As expected, the most common ultrasound signs were vertical artifacts and white lung. A quantitative LUS score for lung aeration assessment has been proposed, based on the identification of four patterns in number and type of visualized artifacts: normal aeration, moderate, severe, and complete loss of aeration [51]. Baldi et al. [52] reported a relationship between the number of B-lines and lung density in mechanically ventilated patients. In patients with VAP, changes in LUS score before and after antibiotics predicted improvement in lung aeration. These findings are consistent with the fact that CT scan computes lung density, which is also the main determinant of appearance, number, and coalescence of LUS artifacts. 

In agreement with recent evidence that connects the vertical artifacts to acoustic traps that can capture acoustic energy and restore it over time, and white lung with a relatively random scattered distribution, a relationship between CT and US features can be speculated [53]; for example, between septal enlargement and vertical artifacts, or between ground glass opacity in CT and white lung. Equally probable is a deteriorating progression, in terms of ventilation, from less dense vertical artifacts to white lung and consolidations.

Consolidations were significantly more frequent in severe and critical patients. In consolidations, the alveoli are filled by inflammatory exudation, and/or collapsed. If the role of consolidations in causing a shunt effect is recognized, it can be speculated that consolidations are aggravating factors and indicators of cytokine storm, vascular damage, ARDS, or bacterial superinfection.

As regards the laboratory indicator, we found a difference between the ordinary and severe and critical phenotypes. The decrease of lymphocytes in the severe and critical patients indicated that many immune cells had been consumed, and the immune function was inhibited. Damage to lymphocytes may be critical in-patient exacerbation, and the decreased lymphocytes could be used as an important index in the evaluation of disease severity. The increased values of neutrophil ratio, C-reactive protein, and procalcitonin in severe and critical patients may have been related to cytokine storm induced by virus invasion, and to comorbidity with other kinds of infections, which has been supported by recent studies [29]. The timely prevention of infection may help reduce complications and mortality.

The strength of this study concerns the robustness of the data obtained from a large, real-life, general adult population, where COVID-19-positive individuals with pneumonia were included. Moreover, all patients were evaluated by the same CT and US methodology, including the score used.

However, the study has some limitations. Ultrasound scans were performed by different operators and the inter-operator agreement was not assessed, due to the technical difficulties imposed by the emergency clinical context. 

Moreover, in our study there was no control group, as its purpose was not to validate the test nor to provide a differential diagnosis with other diseases based on ultrasound imaging, but to evaluate how useful ultrasound can be in the management of a COVID setting. 

It is also important to consider that our results were linked to the prevalence of the disease, which was particularly high during the first wave of the pandemic. With a lower prevalence, the significance of our analysis should be revisited.

## 5. Conclusions

LUS score was independently able to predict a higher risk of adverse events in patients with COVID-19. Indeed, patients with higher LUS scores were more likely to have higher levels of cardiac injury, coagulopathy, and inflammatory biomarkers, more non-invasive ventilation with c-pap or b-level, higher incidence of respiratory failure, ARDS, and sepsis, and higher mortality. The US does not replace the CT examination gold standard for interstitial pneumonia, and US has poor capacity for risk stratification compared to CT. More interesting is the role of US for monitoring patients at the bedside every day during hospitalization, for clinical and instrumental correlation, for follow-up, and for reducing the risk of radiation. Likewise, chest CT has played a crucial role in characterizing pulmonary lesions during the COVID-19 pandemic. It can accurately evaluate the type and extent of lung lesions and could be used to evaluate the severity of the disease. LUS allows only a superficial mapping of the lung, and the LUS score in theory is not perfectly comparable to the CT score (a volumetric estimate of injury). It is conceivable that a certain congruence of results is linked to the superficial expression of lesions, and that therefore in COVID-19 relative symmetry between volume and surface of the pulmonary lesions is maintained. In conclusion, LUS and chest CT have been shown to play a synergistic role in the diagnosis and severity evaluation of COVID-19.

## Figures and Tables

**Figure 1 diagnostics-12-01856-f001:**
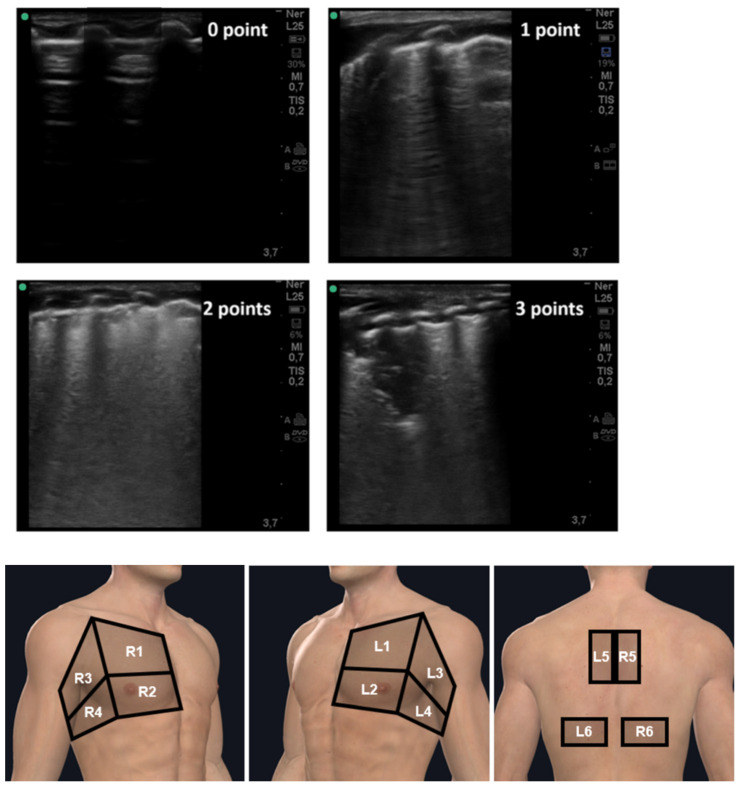
12-zone method; anterior, lateral, and posterior chest. In each zone a score was assigned; 0 = no B-lines; 1 = multiple spaced or isolated B-lines; 2 = diffused coalescent B-lines; 3 = lung consolidation.

**Figure 2 diagnostics-12-01856-f002:**
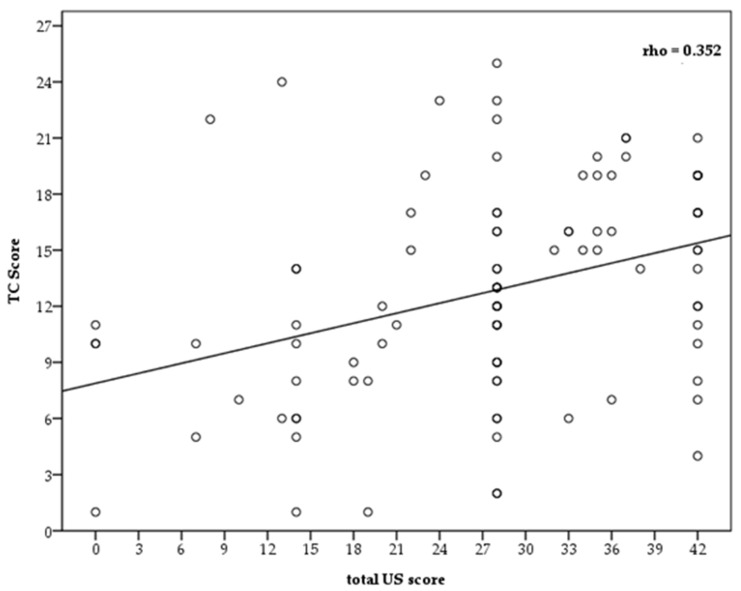
Linear regression analysis and dispersion graph.

**Figure 3 diagnostics-12-01856-f003:**
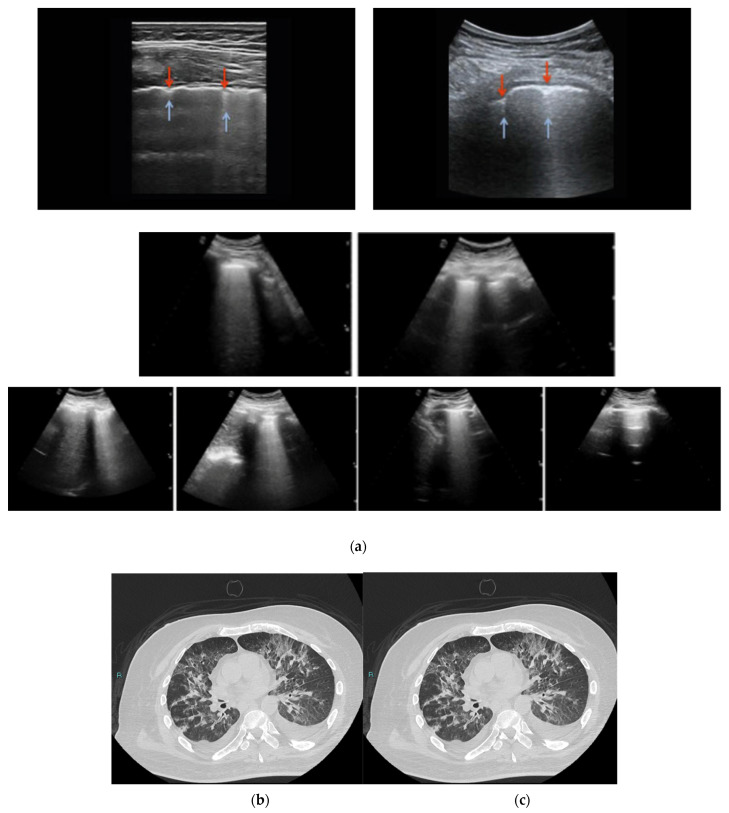
CT and LUS imaging scans of different scores. In these figures the pleural line (indicated by red arrows) is indented, and vertical areas of white (blue arrows) are visible below the indent, which reflect local alterations in the acoustical properties of the lung caused by replacement of air with water, blood, or collapsed tissue. A typical case of a COVID-19 pneumonia patient. (**a**) B-lines at the right and left of the lower lateral lung reflecting pneumonia (score 2); (**b**) B-lines at the right and left of the lower lateral lung reflecting pneumonia (score 3); (**c**) chest CT showing multiple infiltrations.

**Table 1 diagnostics-12-01856-t001:** Anthropometric, demographic, clinical, and biochemical characteristics of the study cohort (*n* = 99).

Parameters	Values
Age (years), median [IQR]	71 [58–78]
Sex, *n* (%) M F	68 (68.7) 31 (31.3)
**Signs and Symptoms**	
Presence of fever, *n* (%)	90 (90.9)
Presence of cough, *n* (%)	90 (90.9)
Pharyngeal hyperemia, *n* (%)	63 (63.6)
Asthenia, *n* (%)	76 (76.8)
Vomiting, *n* (%)	5 (5.1)
Diarrhea, *n* (%)	11 (11.1)
Dyspnea, *n* (%)	89 (89.9)
Tachycardia, *n* (%)	67 (67.7)
**Phenotype, *n* (%)** type 1 type 2 type 3 type 4	7 (7.1) 21 (21.2) 60 (60.6) 11 (11.1)
**Pre-existing Comorbidities**	**Values**
Hypertension, *n* (%)	85 (85.9)
Diabetes, *n* (%)	34 (34.3)
Atrial fibrillation, *n* (%)	13 (13.1)
Ischemic heart disease, *n* (%)	33 (33.3)
Ictus, *n* (%)	14 (14.1)
Dementia, *n* (%)	31 (31.3)
Chronic obstructive pulmonary disease (COPD), *n* (%)	56 (56.6)
Active cancer in the last five years, n (%)	14 (14.1)
Hyperinflammatory syndrome %	84 (84.8)
Smoke, *n* (%)	66 (66.7)
**Obesity, *n* (%)** No Grade I Grade II Grade III	58 (59.6) 30 (30.3) 8 (8.1) 2 (2)
Chronic liver disease, *n* (%)	7 (7.1)
Chronic kidney disease, *n* (%)	31 (31.3)
Mortality, *n* (%)	21 (21.2)
Days of hospitalization, median [IQR]	30 [20–40]
Ventilation, median [IQR]	0.50 [0.28–0.60]
**Oxygen interface, *n* (%)**	
None Nasal cannula Venturi Mask CPAP/Bi-level Orotracheal Intubation	14 (14.1) 18 (18.2) 40 (40.4) 25 (25.3) 2 (2)
**Therapy**	
Anticoagulants, *n* (%)	72 (72.7)
Antiplatelets, *n* (%)	35 (35.4)
ACE Inhibitors, *n* (%)	32 (32.3)
Ultrasound thoracic, median [IQR]	28 [20–36]
**Ultrasound score, *n* (%)**	
Score 0	2 (2)
Score 1	18 (18.2)
Score 2	47 (47.6)
Score 3	32 (32.2)
**CT score, mean (SD)**	12.63 (5.72)
CT score, *n* (%) Score 1 Score 2 Score 3	20 (20.2) 59 (59.6) 20 (20.2)

Abbreviations: IQR: interquartile range; SD: standard deviation; M: male; F: female; BMI: body mass index. The biochemical parameters and the blood-gas analysis results are shown in Table 2. In general, patients were characterized by hyper-inflammation syndrome, lymphopenia, high levels of C-reactive protein, neutrophils, and ferritin, and abnormal coagulation parameters (fibrinogen, d-dimer).

**Table 2 diagnostics-12-01856-t002:** Laboratory characteristics of the study population (*n* = 99).

Laboratory	Values
Hb (mg/dL), mean (SD)	12.2 (2.2)
White blood cells (×10^3^), mean (SD)	9.73 (4.38)
Lymphocytes (a.v.), median [IQR]	0.8 [0.6–1.3]
Neutrophils (a.v.), mean (SD)	7.56 (3.15)
Platelets, mean (SD)	282,098 (141,397)
Azotemia (mg/dL), mean (SD)	54.7 (34.85)
Creatinine (mg/dL), median [IQR]	0.9 [0.8–1.2]
Sodium (mmol/L), mean (SD)	138.8 (4.2)
Potassium (mmol/L), mean (SD)	4.8 (3.9)
AST (U/L), mean (SD)	42.6 (124.6)
ALT (U/L), mean (SD)	48.8 (136.8)
Glycemia (mg/dL), median [IQR]	110 [88.3–175]
CRP (mg/dL), median [IQR]	5 [2.6–12]
INR, median [IQR]	1.12 [1.10–1.20]
aPTT (s), mean (SD)	31 (6.8)
Fibrinogen (mg/dL), mean (SD)	483.3 (141.4)
Nt-pro-bnp (pg/mL), median [IQR]	1578 [600–3500]
D-Dimer (pg/mL), median [IQR]	2300 [782.5–4210]
LDH (mU/mL), mean (SD)	361.5 (138.4)
Troponin (ng/mL), median [IQR]	0.032 [0.014–0.090]
Procalcitonin (ng/mL), median [IQR]	0.2 [0.03–0.90]
Ferritin (ng/mL), median [IQR]	450 [280–700]
**Blood Gas Analysis**	
pH, median [IQR]	7.45 [7.40–7.47]
pO_2_ (mmHg), median [IQR]	68 [58.3–84.8]
pCO_2_ (mmHg), median [IQR]	35 [33–42]
HCO^3-^ (mmol/L), median [IQR]	25 [23–26]
spO_2_ (%), median [IQR]	93.1 [90–96]
Lactates (mmol/L), median [IQR]	2.25 (1.02)
P/F, median [IQR]	231 [136.3–295.3]
FiO_2_ admission, median [IQR]	0.30 [0.21–0.50]

Abbreviations: IQR: interquartile range; SD: standard deviation; Hb: hemoglobin; PLT: platelets; AST: aspartate aminotransferase; ALT: alanine aminotransferase; CRP: C-reactive protein; LDH: lactate dehydrogenase; CPK: creatine phosphokinase. Reference ranges: [Hb] F = 12–16/M = 12–18 g/dL; WBCs: 4500–11,000, Neutrophils: 1500–7000; Lymphocytes: 1500–7000; PLT: 150,000–450,000; Azotemia: 15–50 mg/dL; Serum Creatinine: 0.51–0.95 mg/dL; Sodium: 135–145 mmol/L; Potassium: 3.5–5 mEq/L; AST (F = 8–43 U/L; M = 8–48 U/L); ALT (F = 7–45 U/L, M = 7–55 U/L); Glycemia: 60–110 mg/dL; CRP: 5–10 mg/dL; INR: 0.9–1.3; aPTT: 28–40 s; Fibrinogen: 200–400 mg/dL; NT-proBNP: ≤900 pg/mL; D-Dimer: <500 pg/mL; LDH: 80–300 mU/mL; Troponin: <0.1; procalcitonin: 0–1; ferritin: M: 20–200 ng/mL, F: 20–120 ng/mL; Iron: M: 31–144 μg/dL, F: 25–156 μg/dL; CPK: 60–190 U/L. Blood gas ranges: pH: 7.35–7.45; pO_2_: 80–100 mmHg; pCO_2_: 35–45 mmHg; HCO^3−^: 22–26 mmol/L; spO_2_: 95–100%; Lactates: <2 mmol/L.

**Table 3 diagnostics-12-01856-t003:** Univariate analysis of relationships between ultrasound thoracic score and other parameters in patients infected by COVID-19.

Parameters	Correlation Coefficient	*p*
Age (years)	0.289	0.034
Dyspnoea	0.319	0.051
Tachycardia	0.457	0.002
COVID-19 phenotype	0.589	<0.001
Dementia	0.197	0.256
Platelets	−0.118	0.286
Prothrombin time	0.057	0.917
NT-proBNP	0.174	0.419
D-dimer	0.218	0.047
pH	−0.469	0.008
pO_2_	−0.486	0.003
spO_2_	−0.226	0.467
P/F	−0.689	<0.001
Death of patients	0.492	0.008
Ventilation	0.562	<0.001
Lactates	0.479	0.001

**Table 4 diagnostics-12-01856-t004:** Univariate analysis of the relationships between CT score and other parameters in patients infected by COVID-19.

Parameters	Correlation Coefficient	*p*
Age (years)	0.369	0.029
Dyspnoea	0.488	<0.001
Tachycardia	0.321	0.007
COVID-19 phenotype	0.639	<0.001
Dementia	0.124	0.298
Platelets	−0.189	0.321
Prothrombin time	0.025	0.874
NT-proBNP	0.098	0.513
D-dimer	0.289	0.041
pH	−0.396	0.019
pO_2_	−0.470	<0.001
spO_2_	−0.199	0.148
P/F	−0.762	<0.001
Death of patients	0.466	0.001
Ventilation	0.503	<0.001
Lactates	0.442	0.001

**Table 5 diagnostics-12-01856-t005:** ROC-AUC value of the LUS score compared to the different factor.

AUC of LUS Score	AUC (95% CI)
Dyspnea	0.77 (0.61–0.92)
Tachycardia	0.79 (0.68–0.879
Dementia	0.57 (0.45–0.69)
Mortality	0.70 (0.59–0.82)

## Data Availability

The raw data supporting this study will be made available by the authors if requested.

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
