# Peer review of "The Role of Lung Ultrasound in SARS-CoV-19 Pneumonia Management"

_diagnostics, 2022, doi:10.3390/diagnostics12081856_

Round 1

Reviewer 1 Report

1) Abstract. Results: Ninety-nine patients were included in the study (male 31 68.7%, median age 71). 40.4% of patients required a Venturi mask and 25.3% non-invasive ventila- 32 tion (C-PAP/Bi-level). The overall mortality rate was 21.2% (median hospitalization 30 days). The 33 median ultrasound thoracic score was 28 (IQR 20-36). As for the CT evaluation, the mean score was 34 12.63 (SD 5.72), with most of the patients have lus score patterns 2 (59.6%). The bivariate correlation 35 analysis displayed statistically significant and high positive correlations between both the CT and 36 composite LUS score and ventilation, lactates, COVID-19 phenotype, tachycardia, dyspnea and 37 mortality. Moreover, the most relevant and clinically important inverse proportionality regards P/F, 38 i.e., a decrease in P/F levels was indicative of higher levels of the LUS/CT score. Could you please underline the most important statistical data to support the results?

2) Introduction. L45-50. Severe Acute Respiratory Syndrome is the major complication of the Coronavirus  Disease 2019 infection (SARS-COV–19) [1]. Its clinical spectrum is broad, including  asymptomatic infection, flu-like illness, and severe and diffuse lung involvement, with a  life threating course. Cytopathic and immune-mediated injury in the pulmonary parenchyma are the most significant aspects of this disease [2,3]. Thrombotic microangiopathy  and complement activation, pulmonary embolism and elevated D-dimer levels have also  been reported with a high frequency in patients with COVID-19 [4-7]. Please improve this paragraph and add this reference:

a- Systemic Inflammatory Predictors of In-Hospital Mortality in COVID-19 Patients: A Retrospective Study. Diagnostics 202212, 859. https://doi.org/10.3390/diagnostics12040859

3) Lung ultrasound (LUS) shows good sensitivity and some advantages for detecting 65 and monitoring COVID-19 "pneumonia". It is safe, repeatable, and can be used at the bed- 66 side in the absence of radiation exposure [17,18]. Please improve this sentencence and add these references:

a-The role of ultrasound lung artifacts in the diagnosis of respiratory diseases. Expert Rev Respir Med. 2019;13(2):163-172. doi:10.1080/17476348.2019.1565997

b-  High-Resolution Computed Tomography and Lung Ultrasound in Patients with Systemic Sclerosis: Which One to Choose?. Diagnostics (Basel). 2021;11(12):2293. Published 2021 Dec 7. doi:10.3390/diagnostics11122293

4) 3. Results L227 Ninety-nine patients were included in the study, with a median age of 71 years (IQR 228 58-78 yrs.). 68.7% were male. Please underline in the text the most important statistical values to support the results.

5) Table 3. Univariate analysis regarding the relationship between Ultrasound thoracic score and other parameters in patients infected by COVID-19.  Please add the most important r-values.

6) 4. Discussion L311-316. Lung CT is currently the standard against which other imaging methods for the anatomical definition down to the level of the secondary lung lobule is compared. The most important radiology societies [11–16] recommend the use of CT in the presence of moderate and severe features of COVID-19 when RT-PCR results are negative or  not available, when there is high pre- test probability, and in the management of patients  with worsening or severe respiratory symptoms.

7) 4. Discussion  In this study, we found that the most common CT findings were GGO, consolidation 403 and crazy-paving pattern, including “spider web sign” (defined as a triangular or angular 404 GGO in the subpleural lung with the internal interlobular septa thickened like a net). As 405 expected, the most common ultrasound signs were vertical artifacts and White lung [35]. Please improve this part of discussion and add these references:

a- The role of chest CT in deciphering interstitial lung involvement: systemic sclerosis versus COVID-19. Rheumatology (Oxford). 2022;61(4):1600-1609. doi:10.1093/rheumatology/keab615

b-Different Methods to Improve the Monitoring of Noninvasive Respiratory Support of Patients with Severe Pneumonia/ARDS Due to COVID-19: An Update. J Clin Med. 2022;11(6):1704. Published 2022 Mar 19. doi:10.3390/jcm11061704

8) 5. Conclusions L441-447. LUS is a portable and repeatable method for detecting superficial densities in the  explorable lung. It is a low-cost imaging technique which can be quickly performed at the  bedside by every expert physician. Anatomic alterations of COVID-19 are easily explored  by ultrasound. In the present study the LUS score was sufficiently comparable with a CT  score and was able to predict a higher risk of adverse events in patients with COVID-19. The baseline LUS score (within 24 h of admission) could be roughly linked to the eventual  need for invasive mechanical ventilation and could be a strong predictor of mortality. Could you please underline the novelty of the study and the clinical implications?

Author Response

Dear Editor,

Dear Reviewers,

Thank you very much for the revision of the above manuscript.

We appreciated the further criticisms of the reviewers and we addressed all points raised revising the manuscript accordingly, which resulted in an improved version.

We are hereby resubmitting it along with a point-by-point reply.

We hope that the revised manuscript is now suitable for publication in Diagnostics

Looking forward to hearing from you,

Yours sincerely,

On behalf of all Authors

Marina Lugarà

Point by point response

Reviewer 1: 

1 ) Abstract. Results: Ninety-nine patients were included in the study (male 31 68.7%, median age 71). 40.4% of patients required a Venturi mask and 25.3% non-invasive ventilation (C-PAP/Bi-level). The overall mortality rate was 21.2% (median hospitalization 30 days). The median ultrasound thoracic score was 28 (IQR 20-36). As for the CT evaluation, the mean score was 12.63 (SD 5.72), with most of the patients have lus score patterns 2 (59.6%). The bivariate correlation analysis displayed statistically significant and high positive correlations between both the CT and composite LUS score and ventilation, lactates, COVID-19 phenotype, tachycardia, dyspnea and mortality. Moreover, the most relevant and clinically important inverse proportionality regards P/F,  i.e., a decrease in P/F levels was indicative of higher levels of the LUS/CT score. Could you please underline the most important statistical data to support the results?

RE:

Inverse proportionality P/F levels and LUS and TC score was evaluated by  univariate analysis

P/F-TC score correlation  coefficient  -0,762  p<0,001 (tab 4)

 P/F -LUS score correlation coefficient -0,689 p>0,001 (tab 3)

2   Introduction.  Severe Acute Respiratory Syndrome is the major complication of the Coronavirus Disease 2019 infection (SARS-COV–19) [1]. Its clinical spectrum is broad, including asymptomatic infection, flu-like illness, and severe and diffuse lung involvement, with a life threating course. Cytopathic and immune-mediated injury in the pulmonary parenchyma are the most significant aspects of this disease [2,3]. Thrombotic microangiopathy and complement activation, pulmonary embolism and elevated D-dimer levels have also been reported with a high frequency in patients with COVID-19 [4-7]. Please improve this paragraph and add this reference:

3 : Lung ultrasound (LUS) shows good sensitivity and some advantages for detecting 65 and monitoring COVID-19 "pneumonia". It is safe, repeatable, and can be used at the bed- 66 side in the absence of radiation exposure [17,18]. Please improve this sentence and add these references:

RE : questions 2-3

COVID-19 is an infectious disease with a wide range of clinical symptoms, from asymptomatic to mildly symptomatic and severe forms, pointing to a major role of the host response to SARS-CoV-2 (severe acute respiratory syndrome coronavirus 2)[1]. The clinical spectrum of SARS-CoV-2 infection is broad, ranging from asymptomatic infection to flu-like illness to severe and diffuse viral pneumonia with a life threating course, related to cytopathic and immune -mediated injury in the pulmonary parenchymal .Patients may show the following symptoms: fever, high temperature, cough, myalgia, sputum production, headache, hemoptysis, diarrhea, dyspnea, and, in some cases, acute respiratory distress syndrome (ARDS), acute cardiac injury, or secondary infection.  Most of the infections are not severe, but 81% are mild, 14% of the cases are severe (with dyspnea, hypoxia, or >50% lung involvement on diagnostic imaging), and 5% develop a critical disease with respiratory failure, shock, or multiorgan dysfunction [2]. The risk of death from COVID-19 strongly depends on the age and previous health status. Older patients are much more prone to critical and fatal disease outcomes, especially with comorbidities, such as cardiovascular diseases, hypertension, chronic kidney disease, diabetes, and pulmonary disease [3). Thrombotic microangiopathy and complement activation, pulmonary embolism and elevated D-dimer levels have also been reported with high frequency in patients with COVID-19 [4-7]. Numerous previous studies, including the paper by Giannini and al. [8-9], have already discussed the significance of the D-dimer level as an independent predictor of mortality in severe cases of ARDS during SARS-CoV-2. COVID-19 laboratory diagnosis is based on real time polymerase chain reaction (RT-PCR) assay obtained by oro-nasopharyngeal swab sample, bronchoalveolar lavage or tracheal aspirate, whereas imaging plays a major role in the early diagnosis of the pleuropulmonary complications [8,9-10]. The pathophysiology of severe COVID-19 infection is marked by elevated numbers of neutrophils in the nasopharyngeal epithelium, distal parts of the lungs, and in blood. The experience gained during the Italian epidemic pointed to the patients’ age as one of the most important risk factors for COVID-19 mortality [11]. However, a recent study demonstrated that patients who died of COVID-19 appear to have lost considerable lifetime, independent of their age. Imaging findings significantly support clinical judgement ensuring effective and timely management and in prognosis; indeed, the identification of disease severity allows an appropriate selection of early involvement of the intensive care [10-12]. Contrary to X-ray, Chest Computed tomography (CT) plays a pivotal role in the diagnosis and monitoring of interstitial pneumonia [13,14]. Typical CT patterns of Covid related pneumonia is multifocal bilateral peripheral ground glass opacities associated with subsegmental patchy consolidations, commonly subpleural and predominantly involving lower lung lobes and posterior segments [15,16]. Likewise, lung ultrasound (LUS) shows some advantages for detecting and monitoring COVID-19 "pneumonia"[17] This diagnostic technique is safe, repeatable, and can be used with low cost and bedside in absence of radiation exposure [16-17-,18]. Moreover, it is useful for rapid assessment of the severity of Sars-Cov2 pneumonia/ARDS (acute respiratory distress syndrome) in the diagnosis and follow-up settings, and for monitoring lung in recruitment maneuvers and during prone positioning [19,20]. The use of LUS for patient with suspected COVID-19 may reduce the risk associated with transporting unstable patients to CT rooms, especially important for preventing nosocomial outbreaks due to high contagiousness of virus [21,22] Thoracic ultrasound is employed for the diagnosis of many thoracic diseases and is an accepted detection tool of pleural effusions, atelectasis, pneumothorax, and pneumonia. However, the use of ultrasound for the evaluation of parenchymal lung disease, when the organ is still aerated, is a relatively new application. The diagnosis of a normal lung and the differentiation between a normally aerated lung and a lung with interstitial pathology is based on the interpretation of ultrasound artifacts universally known as A and B-Lines. Even though the practical role of lung ultrasound artifacts is accepted by many clinicians, their physical basis and the correlations between these signs and the causal pathology is not known in depth.[23]. The utility of a lung ultrasound (LUS) in the diagnosis of ILD in very early SSc has also been described and, more recently, its potential for the detection of SSc-ILD in asymptomatic preclinical stages. Recent research has focused on the predictive value of LUS [24-25], which is promising for the application of LUS as a screening method for SSc-ILD in clinical practice. Although these are strong arguments in favor of the application of LUS in SSc, to date, there is no unanimous consensus as to the role LUS plays in the diagnosis and/or prognosis of SSc-ILD.[26] The use of LUS for patient with suspected COVID-19 may reduce the risk associated with transporting unstable patients to CT rooms, especially important for preventing nosocomial outbreaks due to high contagiousness of virus. The purpose of our study was to determine the role of LUS in the diagnosis and prognosis of SARS-COV 2 pneumonia, considering high-resolution computed tomography (HRTC) as the gold standard

4) 3. Results

Ninety-nine patients were included in the study, with a median age of 71 years (IQR 228 58-78 yrs.). 68.7% were male. Please underline in the text the most important statistical values to support the results.

RE:

Ninety-nine patients were included in the study, with a median age of 71 years (IQR 58-78 yrs.), the incidence of sars-cov2 is higher in male 68,7% than in female 31,3% (tab 1).

5) Table 3. Univariate analysis regarding the relationship between Ultrasound thoracic score and other parameters in patients infected by COVID-19.  Please add the most important r-values.

RE : Both the LUS and CT score were analyzed for correlation with the mortality outcome, coagulation and respiratory parameters using the Spearman correlation coefficient. As reported in Tables 3 and 4, the results were almost similar for both scores. Table 3 showed negative correlation by P/F and LUS score ( r-0,689) , and po2 -LUS score( r -0,486), positive correlation type of ventilation-LUS Score (r 0,562) and lactates-LUS Score (r 0,479) . Table 4 showed negative correlation by P/F -TC score (r -0,689) and po2 (r-0,470) and positive correlation by type of ventilation (0,530).

6-7) 4. Discussion L311-316. Lung CT is currently the standard against which other imaging methods for the anatomical definition down to the level of the secondary lung lobule is compared. The most important radiology societies [11–16] recommend the use of CT in the presence of moderate and severe features of COVID-19 when RT-PCR results are negative or  not available, when there is high pre- test probability, and in the management of patients  with worsening or severe respiratory symptoms.

  1. Discussion  In this study, we found that the most common CT findings were GGO, consolidation 403 and crazy-paving pattern, including “spider web sign” (defined as a triangular or angular 404 GGO in the subpleural lung with the internal interlobular septa thickened like a net). As 405 expected, the most common ultrasound signs were vertical artifacts and White lung [35]. Please improve this part of discussion and add these references:

As  expected, the most common ultrasound signs were vertical artifacts and White lung [35]. Please improve this part of discussion and add these references:

a- The role of chest CT in deciphering interstitial lung involvement: systemic sclerosis versus COVID-19. Rheumatology (Oxford). 2022;61(4):1600-1609. doi:10.1093/rheumatology/keab615

b-Different Methods to Improve the Monitoring of Noninvasive Respiratory Support of Patients with Severe Pneumonia/ARDS Due to COVID-19: An Update. J Clin Med. 2022;11(6):1704. Published 2022 Mar 19. doi:10.3390/jcm11061704

RE:  questions 6-7

A differential diagnosis between COVID 19 and systemic sclerosis (SSc) with interstitial lung is possible employing the CT images ; the presence of consolidations and fibrosis inside GGO in the lower lobes are independent CT diagnostic feature for COVID 19 [50]. As expected, the most common ultrasound signs were vertical artifacts and White lung. A quantitative LUS score has been proposed for lung aeration assessment, based on the identification of four patterns in function of number and type of visualized artifacts: normal aeration, moderate, severe, and complete loss of aeration [51]. Baldi et al [52] reported a relationship between the number of B lines and lung density in mechanically ventilated patients. In patients with VAP, changes in LUS score before and after antibiotics predicted the improvement in lung aeration. These findings are consistent with the fact that CT scan computes the lung density, which is also the main determinant of appearance, number, and coalescence of LUS artifacts In agreement with the recent evidence that connects the vertical artifacts to acoustic traps which can capture acoustic energy and restoring it over time, and the white lung with a relatively random scatterer distribution, a relation between CT and US features can be speculated [53], for example the relations between septal enlargement and vertical artifacts and between ground glass opacity in CT with white lung. Equally probable is a damaging progression, in terms of ventilation, from less dense vertical artifacts to white lungs and consolidations.

Consolidations were significantly more frequent in severe/critical patients. In consolidations the alveoli are filled by inflammatory exudation, and/or collapsed. If the role of consolidations in causing a shunt effect is known, it can be speculated that consolidations are aggravating factors as indicators of cytokine storm, vascular damage, ARDS or bacterial superinfection. As regards the laboratory indicator, we found a difference between the ordinary and severe/critical phenotypes. The decrease of lymphocytes in the severe/critical patients indicates that many immune cells are consumed, and the immune function is inhibited. Damage to lymphocytes may be critical in the exacerbations of patients, and the decreased lymphocytes could be used as an important index in the evaluation of disease severity. The increased values of neutrophil ratio, C-reactive protein, and procalcitonin in severe/critical patients may be related to cytokine storm induced by virus invasion and to being comorbid with other kinds of infections, which was supported by recent studies [29]. The timely prevention of infection may help reduce complications and mortality. The strength of this study concerns the robustness of the data obtained from a large, real-life, general adult population, where COVID-19 individuals with pneumonia were included. Moreover, all patients were evaluated by the same CT and US methodology, including the score used.  However, it has some limitations. Ultrasound scans were performed by different operators and the inter operator agreement was not assessed due to the technical difficulties imposed by the emergency clinical context. Moreover, in our study there was no control group, as its purpose was not to validate the test or provide a differential diagnosis with other diseases based on ultrasound imaging, but to evaluate how useful ultrasound can be in the management of a Covid setting. It is also important to consider that our results are linked to the prevalence of the disease, particularly high during the first wave of the pandemic. With a lower prevalence, the significance of our analysis needs to be revisited.

  1. Conclusions. LUS is a portable and repeatable method for detecting superficial densities in the explorable lung. It is a low-cost imaging technique which can be quickly performed at the bedside by every expert physician. Anatomic alterations of COVID-19 are easily explored by ultrasound. In the present study the LUS score was sufficiently comparable with a CT score and was able to predict a higher risk of adverse events in patients with COVID-19. The baseline LUS score (within 24 h of admission) could be roughly linked to the eventual need for invasive mechanical ventilation and could be a strong predictor of mortality. Could you please underline the novelty of the study and the clinical implications?

RE:

LUS score was able to predict a higher risk of adverse events in patients with COVID-19 independently. Indeed, patients with the highest LUS score were more likely to have higher levels of cardiac injury, coagulopathy and inflammatory biomarkers, more non-invasive ventilation with c-pap/b-level, higher incidence of respiratory failure, ARDS, sepsis and higher mortality. The US don’t replace the CT examination gold standard for interstitial Pneumonia and Us has poor role for risk stratification compared to CT . More interesting the role US for monitoring patient bed side every day during hospitalization, for clinical and instrumental correlation, for follow-up and for reduce the risk of radiation. Likewise, Chest CT has played a crucial role in characterizing pulmonary lesions during the COVID-19 pandemic. It can accurately evaluate the type and extent of lung lesions and could evaluate the severity of the disease. LUS allows only a superficial mapping of the lung, and the LUS score in theory is not perfectly comparable to the CT score (a volumetric estimate of injury).  It is conceivable that a certain congruence of results is linked to the superficial expression of the lesions, therefore in COVID-19 relative symmetry between volume and surface of the pulmonary lesions is maintained. In conclusion, LUS and Chest CT have shown a synergistic role in the diagnosis and disease severity evaluation of Covid 19.

Hoping our answer are satisfactory.

     Kind regards,

Marina Lugarà

Reviewer 2 Report

Thank you for the opportunity to review your manuscript. I consider that your manuscript is of interest as it helps define better the diagnostic value of LUS in COVID-19. 

Several authors report that LUS can diagnose with good/high accuracy several cardiorespiratory conditions, such as pulmonary oedema, pleural effusion, pneumothorax, pulmonary embolism, atelectasis, pneumonia, and ARDS. Moreover, some authors report that LUS findings have a prognostic value in patients with ALI/ARDS (Zhao et al., 2015, Li et al., 2022). However, there is increased variability in the methods used in the studies aiming to assess LUS diagnostic or prognostic value, with several scanning protocols and scoring systems being described. What scanning protocol has the highest diagnostic value in a specific clinical context? What scoring system? These are still open questions that need to be clarified. Thus, I consider that this manuscript can be of interest as it reports the significance of one of the previously described LUS protocols in the diagnosis of COVID-19.   

Author Response

Dear Editor,

Dear Reviewers,

Thank you very much for the revision of the above manuscript.

We appreciated the further criticisms of the reviewers and we addressed all points raised revising the manuscript accordingly, which resulted in an improved version.

We are hereby resubmitting it along with a point-by-point reply.

We hope that the revised manuscript is now suitable for publication in Diagnostics

Looking forward to hearing from you,

Yours sincerely,

On behalf of all Authors

Marina Lugarà

This manuscript is a resubmission of an earlier submission. The following is a list of the peer review reports and author responses from that submission.

Round 1

Reviewer 1 Report

Thank you for the opportunity to review Your manuscript. I consider Your research interesting and very topical. The objectives are clearly stated. The methods are well described. The results are consistent with the study objectives. In Discussion, study findings are interpreted and well contextualized in the general field. Please consider adding the Ethics Research Committee approval (date and number).

Author Response

Re: The role of the Chest Computed Tomography and Lung ultrasound in the SARS-COV-19 pneumonia management

Dear Editor,

Dear Reviewers,

Thank you very much for the revision of the above manuscript.

We appreciated the further criticisms of the reviewers and we addressed all points raised revising the manuscript accordingly, which resulted in an improved version.

We are hereby resubmitting it along with a point-by-point reply.

We hope that the revised manuscript is now suitable for publication in Diagnostics

Looking forward to hearing from you,

Yours sincerely,

On behalf of all Authors

Marina Lugarà

Point by point response

Reviewer 1

Please consider adding the Ethics Research Committee approval (date and number).

Re:

The ethics research committee was informed of the study on date 28/07/2021 prot. 0179104/i

Reviewer 2 Report

Lugarà and Co-Authors presented a study on the role of two imaging methods, computed tomography (CT) and lung ultrasound (LUS), for the management of SARS-CoV-2 pneumonia.

The Authors reported data on patients enrolled in a single center in Italy, between March and October 2020.

Several data were published on CT scan and LUS in this population of patients.

I have some questions for the Authors.

How they defined the sample size of the study?

It seems a cohort study but no sample size calculation paragraph is reported in the paper.

Moreover, are the patients consecutive enrolled? It seems so but it is not clearly reported and no ‪STARD diagram to report flow of participants is present. These data have to be added.

Other Italians data were already published by enrolling patients in the same period of time (the first pandemic wave in Italy, temporally different than other countries). These studies reported a larger number of participants in a smaller period of time that the 8-month period of the present study. How the Authors can explain these results and how it could have an impact on the generalizability of the study?

During the first wave there was be a so large regional difference in COVID-19 cases in Italy? Could the Authors add reference on this point and adding it in the Discussion? 

The use of LUS by operators with different training (“3 internists and 2 radiologists”) is very interesting. Worldwide, radiologists aren’t used to perform LUS, but unfortunately the Authors reported no inter-raters agreement was possible.

Previous international societies and consensus conference suggested a higher depth for LUS than that used in the study. Why the Authors chose such apparently reduced depth for evaluating lung parenchyma?

The LUS scheme is definitively the most used but I suggest to add a reference about this for facilitating readers.

The TC scans were reported to be performed within 48 hours. How impact this temporal distance could have on the reported results?

This point isn’t discussed in the Limitations section (that seems very concise and should be better defined).

Table 1 reported interesting data but some of them are also reported in the Results paragraph. I suggest to short the table to make it more readable and leave them in the text.

Figure 2 might be added as supplemental materials.

The Authors reported as study aim, “to determine the role of high-resolution computed tomography and LUS in the diagnosis and prognosis of SARS-CoV-2 pneumonia” and as endpoint “to evaluate the usefulness of Chest CT severity score and lung ultrasound score in Covid clinical management. As well, we further assessed the correlation between imaging severity score to oxygen requirement.”

Endpoints on mortality are likely related to prognosis but not reported in the Methods section as well as coagulation and respiratory parameters were explained in such a paragraph.

Methods section seems to be clarified and completed.

Moreover, the reported aim was about diagnosis but a few diagnostic accuracy results are present.

AUC-ROC have to be added in the Methods section as well as the Youden’ index used for defining the better LUS score for defining the outcome. Why the Authors chose this test instead of the Lin’ one or the minimum distance’s test?  Could the Authors add data on clinical usefulness? It seems they have all data available to do that and it is definitely related to the study aim.

In the Results section, Table 3 and 4 are univariate analyses on correlation between LUS or CT and several parameters. Why the Authors chose this type of analysis? If enough degrees of freedom were available, regression models were more appropriate to answer the study aim.

The two table should be put together.

I also strongly suggest to summarize Table 6-11. In the present for there are some repeats between text and tables and they aren’t easy to navigate.

I hope to have the possibility to read the revised for of the manuscript.

Author Response

Comments and Suggestions for Authors
Lugarà and Co-Authors presented a study on the role of two imaging methods, computed tomography (CT) and lung ultrasound (LUS), for the management of SARS-CoV-2 pneumonia.

The Authors reported data on patients enrolled in a single center in Italy, between March and October 2020.

Several data were published on CT scan and LUS in this population of patients.

I have some questions for the Authors.

How they defined the sample size of the study?
It seems a cohort study but no sample size calculation paragraph is reported in the paper.
Moreover, are the patients consecutive enrolled? It seems so but it is not clearly reported and no STARD diagram to report flow of participants is present. RE: The sample size was calculated using one-sample proportion test by STATA 14. For a significance level of 0.05 and a statistical power of 0.8 was needed a minimal sample size of 86 patients.

Other Italians data were already published by enrolling patients in the same period of time (the first pandemic wave in Italy, temporally different than other countries). These studies reported a larger number of participants in a smaller period of time that the 8-month period of the present study. How the Authors can explain these results and how it could have an impact on the generalizability of the study?

During the first wave there was be a so large regional difference in COVID-19 cases in Italy? Could the Authors add reference on this point and adding it in the Discussion?  RE: thank you for your suggestions. All these questions were added in discussion section.

The use of LUS by operators with different training (“3 internists and 2 radiologists”) is very interesting. Worldwide, radiologists aren’t used to perform LUS, but unfortunately the Authors reported no inter-raters agreement was possible.
RE: Thank you for this comment. We discussed this point in the limitation
Previous international societies and consensus conference suggested a higher depth for LUS than that used in the study. Why the Authors chose such apparently reduced depth for evaluating lung parenchyma?
RE: We used the technical approach as suggested by Soldati’s score.

The TC scans were reported to be performed within 48 hours. How impact this temporal distance could have on the reported results?This point isn’t discussed in the Limitations section (that seems very concise and should be better defined).
Re: No impact was reported by the temporal distance in performing CT and LUS

Table 1 reported interesting data but some of them are also reported in the Results paragraph. I suggest to short the table to make it more readable and leave them in the text.

RE: Table 1 was performed by the statistician and we have not modified it in order not to alter the basic protocol of the work

The Authors reported as study aim, “to determine the role of high-resolution computed tomography and LUS in the diagnosis and prognosis of SARS-CoV-2 pneumonia” and as endpoint “to evaluate the usefulness of Chest CT severity score and lung ultrasound score in Covid clinical management. As well, we further assessed the correlation between imaging severity score to oxygen requirement.”
Endpoints on mortality are likely related to prognosis but not reported in the Methods section as well as coagulation and respiratory parameters were explained in such a paragraph.
Methods section seems to be clarified and completed. Moreover, the reported aim was about diagnosis but a few diagnostic accuracy results are present.

Re: thank you for these observations. We added these issues in the method section

Why the Authors chose this test instead of the Lin’ one or the minimum distance’s test?  RE: The Youden index was chosen because J point of curve evaluated by this method given equal weight to sensitivity and specificity  

I hope to have the possibility to read the revised for of the manuscript  

Hoping our answer are satisfactory.

     Kind regards,

Marina Lugarà

Reviewer 3 Report

I carefully read and evaluated this manuscript. In my opinion this paper doesn’t provides new insights. There are already many interesting works in literature that support the use of lung ultrasound and its role in patients with interstitial pneumonia secondary to SARS COv2 infection and the possible complementarity and synergistic role between lung ultrasound and chest CT in the diagnosis and assessment of COVID-19 infection. (for example: Yong Yang et al. Intensive Care Med (2020) 46:1761–1763; Soldati G et al. (2020). J Ultrasound. Med. 39, 1413-1419; Perrone T et al. (2020) J. Ultrasound Med. doi:10.1002/jum.15548 28). In addition the ultrasound pictures are of poor quality, often incomprehensible and not adequate to the well known ultrasound imaging found in interstitial pneumonia.

Author Response

Comments and Suggestions for Authors
I carefully read and evaluated this manuscript. In my opinion this paper doesn’t provides new insights. There are already many interesting works in literature that support the use of lung ultrasound and its role in patients with interstitial pneumonia secondary to SARS COv2 infection and the possible complementarity and synergistic role between lung ultrasound and chest CT in the diagnosis and assessment of COVID-19 infection. (for example: Yong Yang et al. Intensive Care Med (2020) 46:1761–1763; Soldati G et al. (2020). J Ultrasound. Med. 39, 1413-1419; Perrone T et al. (2020) J. Ultrasound Med. doi:10.1002/jum.15548 28).

Re: Soldati et al only proposed a method of ultrasound scans with standardized image acquisition protocols, based on a disease severity score. No comparisons were made with CT or clinical data. Moreover, this proposal, made at the beginning of the pandemic, to date has not been validated by a clinical trial with CT. Yong Yang et al wrote a letter comparing ultrasound and CT findings only in terms of sensitivity in a limited set of patients recruited early in the pandemic. The study of Perrone et al. (cited in the discussion section), is very different from ours. It evaluated the potential prognostic value of a LUS protocol in patients with SARS-CoV-2 pneumonia, investigating specifically the association between the LUS score and clinical worsening. No comparison with CT findings was made.

Our work is different. It aims to assess the role of high-resolution computed tomography (HRCT) and lung ultrasound (LUS) in the diagnosis and prognosis of SARS-COV 2 pneumonia. Its material and method section clearly specified the use of a quantitative CT protocol for detecting COVID-19 lesions, compared with an echo image acquisition protocol and with the severity score mentioned in the work on J Ultrasound. Med. 2020, 1413-1419, for the first time subjected to clinical verification. Therefore, we disagree with the Reviewer's conclusions because, in our opinion, the new insights are: 1. A clinical validation of the proposal in J Ultrasound. Med. 39, 1413-1419 and in J Ultrasound Med. 2020 Jul;39(7):1459-1462. 2. A quantitative comparison between CT and US findings. Moreover, both LUS and CT score were further analyzed for potential correlation with mortality outcome and coagulation and respiratory parameters. 3. The robustness of the data obtained from a large, real-life, adult general population, where ill COVID-19 individuals with pneumonia.

In addition the ultrasound pictures are of poor quality, often incomprehensible and not adequate to the well known ultrasound imaging found in interstitial pneumonia.

Re: Thank you for this comment. We added the suggested references. We improved the quality of the figures and conclusions.

Hoping our answer are satisfactory.

     Kind regards,

Marina Lugarà

Round 2

Reviewer 2 Report

I would thank the Authors for the time and the efforts they used for answering my questions and concerns. 
They have added some limitations in a brief paragraph. I was hoping something more detailed and more important, in several points of the point-by-point 
but most of my questions remained without answers. 
What about the difference with other Italians studies? 
The difference in LUS performers is only mentioned and not discussed. 
The data about impact of temporal distance between LUS and CT aren’t reported. Please add them (how this impact was estimated?)

Table 1 is the about the demographic characteristics of patients. 
What the Authors mean about the impossibility to modify it in order to “not alter the basic protocol of the work”? The Table is still very difficult to read. 

The answer to my question about different scanning method is that the Authors used the “Soldati’s score”. The reference is an interesting position paper of Soldati et al about the need and the possibility to apply LUS evaluation during the pandemic, with explanation on how to do it. 
My question was about difference with the other approaches, tested in terms of reproducibility and diagnostic accuracy, in different studies. Please clarify this point and report evidence on this. 

The sentence added in the statistical analysis paragraph isn’t clear to me. What’s the meaning of “mortality in function as function of LUS score”? 
My questions about difference of diagnostic accuracy and prognostic studies seems to remain without answers. 

I have also asked to detail the sample size. 
I think something is missing in the answer. There are only power and alpha, it isn’t possible to calculate a sample size in this way. The answer has to be added in the manuscript. 

I sorry for my concerns and I really hope to have the possibility to read more comprehensively revised version of the manuscript. 

Author Response

Re: The role of the Chest Computed Tomography and Lung ultrasound in the SARS-COV-19 pneumonia management

Dear Editor,

Dear Reviewers,

Thank you very much for the revision of the above manuscript.

We appreciated the further criticisms of the reviewers and we addressed all points raised revising the manuscript accordingly, which resulted in an improved version.

We are hereby resubmitting it along with a point-by-point reply.

We hope that the revised manuscript is now suitable for publication in Diagnostics

Looking forward to hearing from you,

Yours sincerely,

On behalf of all Authors

Marina Lugarà

Point by point response

Reviewer 2:

The data about impact of temporal distance between LUS and CT aren’t reported. Please add them (how this impact was estimated?)

RE: CT and LUS are not competitive but rather complementary tools that can be used in different settings and timing to answer different clinical questions. CT scan offers a better and comprehensive view of the lung and can also help identify complications such as infarction, embolism, emphysema. Therefore, CT scanning is always helpful in case of sudden worsening of clinical conditions or for an initial assessment of moderate to severe patients, when it is available. Conversely, LUS can be used as a first level exam during the first evaluation in the emergency department or even at home to distinguish low-risk from high-risk patients.

The difference in LUS performers is only mentioned and not discussed. 

RE: every expert sonographer recorded images and clip starting from the right posterior basal regions. A uniform and repeatable report form, for every ultrasound, was used among the different operators (Figure 1 in the text) .

Table 1 is the about the demographic characteristics of patients. 

RE: thank you for your suggestion, we have proceeded with the modification of table 1.

I have also asked to detail the sample size. 

RE: The simple size was 79 patients for a confidence interval of 95%

My questions about difference of diagnostic accuracy and prognostic studies seems to remain without answers

 RE:  LUS findings may show good sensitivity and positive predictive values in the context of COVID-19 epidemic (i.e., high “a priori” probability of disease in the presence of respiratory symptoms). In the present study, LUS showed a low specificity (60%). This result is even higher than that reported by a recent meta-analysis by Cochrane of Salameh JP et al, on thoracic imaging tests for the diagnosis of COVID-19, computed for LUS a specificity of 45%. Indeed, ultrasound findings in COVID-19 pneumonia are not specific. According to Sperandeo et al. the specificity of LUS in COVID-19 patients is usually low, since these patients have several co-morbidities such as chronic obstructive pulmonary disease, fibrosis, or heart failure.

My question was about difference with the other approaches, tested in terms of reproducibility and diagnostic accuracy, in different studies. Please clarify this point and report evidence on this. 

 RE:

  In this study we used LUS score (Soldati  et al) , the thorax was scanned in twenty inter-costal zones, six on each hemithorax, depending on patient’s conditions (figures 1 in the text). In critically ill patients, who cannot maintain the sitting position, paravertebral scans could not be acquired, in these cases the operator moved the probe in two directions: as posteriorly as possible to get more posterior information and towards the lowest possible point. A total of 12 regions were assessed using a two-dimensional view with the probe placed perpendicular to the chest wall and evaluated for the following signs: pleural line (a horizontal hyperechoic line between the ribs), A-lines (horizontal reverberation artefacts repeated at a constant distance equal to the distance between pleural line and probe surface), B-lines (vertical hyperechoic artefacts deriving from the pleural line) and consolidation (presence of a tissue-like pattern) ]. Although LUS is not currently considered in the main international guidelines for COVID-19 patient management, some authors have proposed semi-quantitative LUS scores for COVID-19 pneumonia, which can be used to quantify lung aeration. The most widely used is called the lung ultrasound score (LUS), global or total LUS, or global LUS aeration score, in which 0–3 points are assigned for each of the 12 zones according to ultrasound features [32,33,30,34]: 0 = normal; 1 = well- defined B lines (B1); 2 = coalescent B lines or white lung (B2); and 3 = consolidation (Figure 1). Peschel et al. used 0 to 4 points for up to 12 zones; this score, called the lung aeration score (LAS), is assessed by calculating the arithmetic mean of the points of all examined areas. Recently, Zotzmann et al. advocated a combination of LUS and the Wells score as a screening tool for pulmonary embolism (PE) in COVID-19 ARDS.

LUS score is not used for the diagnosis of covi-19 , but can be useful  it for stratification of the risk of death in COVID-19 patients, predicting those who can be at the highest risk of mortality and therefore in need of active and closer surveillance.

Hoping our answer are satisfactory.

     Kind regards,

Marina Lugarà

Reviewer 3 Report

Dear Author,

I carefully read and evaluated the revised manuscript. I am sorry but, in my opinion, the paper doesn’t provides new insights in literature 

Author Response

Thank you suo much for your review, your advice has allowed us to improve our
manuscript.

Round 3

Reviewer 2 Report

I would thank the Authors for their effort. 
Unfortunately, although I would agree with most of the Authors' comments, no answers to my questions related to the present work were reported in the new revision, except for Table 1 that has been modified. 
The sample size calculation was different in Revision 1 and 2 and the question about how it was obtained remained without explanations (i.e. alpha and beta aren't enough to calculate a sample size). 
The same approach was used about diagnosis and prognosis as well as reproducibility and diagnostic accuracy (no answer to my speficic question although I would agree, in general, with Authors' comments but these comments aren't specific for the present study).
I have to admit I would have received some more accurate and detailed answers.